# A Retrospective Propensity Score Matched Analysis Reveals Superiority of Hypothermic Machine Perfusion over Static Cold Storage in Deceased Donor Kidney Transplantation

**DOI:** 10.3390/jcm9072311

**Published:** 2020-07-21

**Authors:** Silvia Gasteiger, Valeria Berchtold, Claudia Bösmüller, Lucie Dostal, Hanno Ulmer, Christina Bogensperger, Thomas Resch, Michael Rudnicki, Hannes Neuwirt, Rupert Oberhuber, Benno Cardini, Stefan Scheidl, Gert Mayer, Dietmar Öfner, Annemarie Weissenbacher, Stefan Schneeberger

**Affiliations:** 1Department of Visceral, Transplant and Thoracic Surgery, Medical University of Innsbruck, 6020 Innsbruck, Austria; silvia.gasteiger@i-med.ac.at (S.G.); valeria.berchtold@i-med.ac.at (V.B.); claudia.boesmueller@tirol-kliniken.at (C.B.); christina.bogensperger@tirol-kliniken.at (C.B.); t.resch@tirol-kliniken.at (T.R.); rupert.oberhuber@i-med.ac.at (R.O.); benno.cardini@i-med.ac.at (B.C.); stefan.scheidl@tirol-kliniken.at (S.S.); dietmar.oefner-velano@tirol-kliniken.at (D.Ö.); 2Department of Medical Statistics, Informatics and Health Economics, Medical University of Innsbruck, 6020 Innsbruck, Austria; Lucie.Dostal@i-med.ac.at (L.D.); hanno.ulmer@i-med.ac.at (H.U.); 3Department of Internal Medicine IV, Nephrology and Hypertension, Medical University of Innsbruck, 6020 Innsbruck, Austria; michael.rudnicki@tirol-kliniken.at (M.R.); hannes.neuwirt@tirol-kliniken.at (H.N.); gert.mayer@tirol-kliniken.at (G.M.)

**Keywords:** kidney transplantation, hypothermic machine perfusion, delayed graft function

## Abstract

Hypothermic machine perfusion (HMP) has been introduced as an alternative to static cold storage (SCS) in kidney transplantation, but its true benefit in the clinical routine remains incompletely understood. The aim of this study was to assess the effect of HMP vs. SCS in kidney transplantation. All kidney transplants performed between 08/2015 and 12/2019 (*n* = 347) were propensity score (PS) matched for cold ischemia time (CIT), extended criteria donor (ECD), gender mismatch, cytomegalovirus (CMV) mismatch, re-transplantation and Eurotransplant (ET) senior program. A total of 103 HMP and 103 SCS instances fitted the matching criteria. Prior to PS matching, the CIT was longer in the HMP group (17.5 h vs. 13.3 h; *p* < 0.001), while the delayed graft function (DGF) rates were 29.8% and 32.3% in HMP and SCS, respectively. In the PS matched groups, the DGF rate was 64.1% in SCS vs. 31.1% following HMP: equivalent to a 51.5% reduction of the DGF rate (OR 0.485, 95% CI 0.318–0.740). DGF was associated with decreased 1- and 3-year graft survival (100% and 96.3% vs. 90.8% and 86.7%, *p* = 0.001 and *p* = 0.008) or a 4.1-fold increased risk of graft failure (HR = 4.108; 95% CI: 1.336–12.631; *p* = 0.014). HMP significantly reduces DGF in kidney transplantation. DGF remains a strong predictor of graft survival.

## 1. Introduction

Kidney transplantation (KTx) is the therapy of choice for patients with end-stage renal disease (ESRD). The shortage of donor organs results in an ever-increasing waiting list and accumulating patient deaths. The donor pool and organ utilization have been expanded in recent years through the use of an increased number of marginal organs [1]. Extended criteria donor (ECD) organs and organs recovered from donation after circulatory death (DCD), however, are more vulnerable to ischemia reperfusion injury. This eventually translates into higher rates of delayed graft function (DGF), primary non-function (PNF) and inferior graft survival [2,3,4,5]. While ECD kidneys have a 1.7 times greater risk for graft failure [6,7], recipients still gain a significant survival benefit compared to dialysis [8]. Nevertheless, ECD organs with prolonged travel time are discarded at a high rate, since transplant surgeons and physicians fear the increased risk of DGF and PNF [1]. Considering the benefit of ECD organ use and further decreasing the discard rate, optimal organ preservation and ex situ organ quality assessment remain key tools for eventually increasing utilization in kidney transplantation. 

Early attempts of dynamic organ preservation in the 1960s [9,10] were not pursued further due to technical and logistical hurdles. The technology was reintroduced decades later and a prospective randomized trial by Moers et al indicated that DGF rates could be lowered by hypothermic machine perfusion (HMP) compared to static cold storage (SCS) [11]. Superior outcomes and/or preferable effects on kidney allografts and transplant outcomes have been confirmed by others [12,13,14]. The positive impact seems to be more pronounced in ECD kidneys and is presumed to result from the protective effect on the endothelium and the removal of waste products [15]. Since the benefit was not profound in some trials and not uniformly considered to outweigh the logistics and costs involved, SCS largely remains the current standard of kidney preservation [16]. The aim of our study was to investigate the impact of HMP on kidney graft function after deceased donor kidney transplantation in an HMP cohort propensity score matched with SCS (PS cohort). 

## 2. Material and Methods

Following institutional ethics board approval (study number 1246/2019), a retrospective analysis of all deceased donor KTx performed at the Department of Visceral, Transplant and Thoracic Surgery, Medical University of Innsbruck, between August 2015 and December 2019, was conducted. Patients had undergone either single or double KTx from deceased donors. The LifePort Kidney Transporter^®^ (Organ Recovery Systems, Itasca, IL, USA) and UW Machine Perfusion Solution (Belzer MPS^®^) were used for HMP. Before undergoing HMP, every kidney experienced a period of SCS. Kidneys from DCD were excluded since they represent only 1% of kidneys in our cohort. En-bloc KTx from pediatric donors, simultaneous pancreas kidney transplants (SPK), combined liver and kidney transplants (LKTx) and combined heart and kidney transplants (HKTx) were also excluded. The study was carried out in accordance with the STROBE (Strengthening The Reporting of OBservational Studies in Epidemiology) checklist [17]. The decision whether a kidney underwent HMP or SCS was based on the availability of immediate operation room (OR) capacity and crossmatch results.

Kidney graft loss was defined as return to chronic dialysis. Graft survival was not censored for death. DGF was defined as the need for at least one dialysis within the first seven days after transplant with the exception of dialysis for hyperkalemia or hypervolemia within the first 12 hours post-transplant [18]. Cytomegalovirus (CMV) mismatch was defined as CMV IgG positive donor to CMV IgG negative recipient (D+R−). “Daytime” procedure was defined as transplantations with skin incision between 8 AM and 8 PM, “Nighttime” transplants were defined by skin incision between 8 PM and 8 AM [19]. ECD was defined as age ≥ 60 years, or age of 50 to 59 years with two of the following: a history of hypertension, a creatinine greater than or equal to 1.5 mg/dl or death resulting from cardiovascular accident as defined in 2002 by Port et al [20].

Standard immunosuppression included induction therapy with 20 mg basiliximab for first-KTx recipients (day 0 and 3); tacrolimus was administered from the first postoperative day onwards, aiming for trough levels of 8–10 ng/mL during the first three months, according to our institutional standard; mycophenolate mofetil 1000 mg was given twice daily together with a steroid taper. Patients undergoing re-transplantation received a single dose of 8 mg/kg antithymocyte globulin (ATG) as induction therapy. For infection prophylaxis, ampicillin/sulbactam was administered for three days. In case of CMV mismatch (D+/R−) or induction therapy with ATG, an antiviral prophylaxis with valganciclovir was applied for 90 days. Patients with DGF received neither additional treatment (i.e. additional induction therapy) nor standardized protocol biopsies.

### Statistical Analysis

Continuous variables are presented as mean ± standard deviation (SD) in case of a normal distribution and as median and range otherwise. Categorical data are presented as number of cases with percentage. 

Baseline characteristics between kidney preservation (HMP/SCS) were tested using the independent-sample Mann–Whitney U test for non-normally distributed continuous variables and *χ^2^*-test or Fisher’s exact test for categorical variables, as appropriate. Standardized mean differences [21] for comparing means and prevalence between groups are reported. 

A propensity score matching with caliper of 0.05 was performed based on following variables: donor type (SCD/ECD), CIT, gender and CMV mismatch, number of transplants (first or Re-Tx) and ET-senior program [22]. The quality of matching was assessed by calculation of the standardized mean difference (SMD) between selected variables, with a SMD < 0.10 reflecting good matching [21].

The effect of HMP on the incidence of DGF was modelled with the conditional logistic regression analysis. Patient and graft survival were calculated with Kaplan–Meier survival analysis method. Kaplan–Meier curves were limited at a number at risk of 10% of patients. The effect of DGF on graft survival was assessed using Cox regression analysis.

All analyses were performed with SPSS (Statistical Package for the Social Sciences) version 25 (IBM; Armonk, NY, USA). A *p*-value of less than 0.05 was considered to indicate statistical significance. 

## 3. Results

In total, 347 patients were transplanted between August 2015 and December 2019. HMP was applied in 124/347 patients (35.7%). In the remaining 223/347 patients (64.3%), kidneys were transplanted following SCS. DGF occurred in 37/124 patients (29.8%) and 72/223 patients (32.3%) following HMP and SCS (*p* = 0.638). Mean CIT was 17.5 ± 5.1 h and 13.3 ± 4.6 h in the HMP and SCS group, respectively (*p* < 0.001, Table 1).

To correct for key donor and recipient factors and CIT, propensity score matching was performed in the ratio of 1:1 (103 HMP kidneys and 103 SCS kidneys). Figure 1 displays the algorithm of patient selection. Following propensity score matching the CIT was 16.3 ± 4.3 and 16.0 ± 4.4 h in the HMP-cohort and SCS-cohort, respectively. Baseline characteristics following PS-matching are shown in Table 2, donor and recipient demographics as well as transplant factors (matched cohort) are shown in Table 3.

### 3.1. Incidence of DGF

DGF occurred in 98/206 patients (47.6%) and was less frequent in recipients of SCD organs (*n* = 101) compared to ECD (*n* = 105); 41/101 (40.6%) vs. 57/105 (54.3%), *p* = 0.049. DGF was significantly less frequent following HMP compared to SCS; (32/103 (31.1%) vs. 66/103 (64.1%), *p* < 0.001, Figure 2A). The difference was significant for both SCD and ECD kidneys: DGF occurred in 13/53 (24.5%) SCD kidneys compared to 19/50 (38%) ECD organs in the HMP group. Following SCS, DGF occurred in 28/48 (58.3%) SCD kidneys, while the DGF rate was 69.1% (38/55) in ECD kidneys; *p* = 0.001 for SCD and ECD (Figure 2B). A conditional logistic regression analysis modelling for the risk of DGF revealed that HMP resulted in a 51.5% reduction of DGF (OR = 0.485; 95% CI: 0.318–0.740; *p* = 0.001). 

### 3.2. Daytime vs. Nighttime Procedures

When stratifying transplant procedures by time of surgery, 39.3% (*n* = 81) of transplants were performed during nighttime. SCS kidneys were more likely to be transplanted during nighttime (56.8% vs. 43.2%, *p* = 0.077) compared to HMP kidneys. Nighttime vs. daytime kidney transplant procedures did not differ in regard to postoperative complication rate (27.2% vs. 27.2%), DGF (49.4% vs. 46.4%) and graft loss (9.5% vs. 8.8%). 

### 3.3. HMP Parameters 

Median HMP time was 6.2 hours (1.9–18.8). Preservation time did not differ between patients with immediate graft function and DGF (6.1 vs. 6.7, *p* = 0.963). The flow increased significantly over the course of HMP (70mL/min (range 6–239) vs. 107mL/min (range 48–240), *p* < 0.001) and the intra-renal-resistance (IRR) index declined over time (0.39 mmHg/mL/min (range 0.6–4.7) vs. 0.22 mmHg/mL/min (range 0.1–0.5), *p* < 0.001). Flow and IRR index at 15 minutes of HMP were 82 mL/min (range 6–239) and 0.30 mmHg/mL/min (range 0.1–4.66) in patients with an immediate graft function compared to 55 mL/min (range 18–191) and 0.46 mmHg/mL/min (range 0.1–1.22) in patients with DGF; *p* = 0.074 and *p* = 0.080, respectively. At the end of perfusion, flow and IRR did not differ among patients with, and without DGF; (95 mL/min vs. 115 mL/min and 0.26 mmHg/mL/min vs. 0.22 mmHg/mL/min, *p* = 0.172 and *p* = 0.130).

### 3.4. Complications

Within the median observational period of 24 months, postoperative complications occurred in 56/206 patients (27.2%) and were comparable in patients receiving kidneys after HMP 27/103 (26.2%) and SCS 29/103 (28.2%). Most common complications were hematoma (*n* = 14/6.8%), lymphoceles (*n* = 19/9.2%), urological complications (*n* = 15/7.3%), wound infections (*n* = 13/6.3%) and vascular complications (*n* = 4/1.9%). Wound infections occurred significantly more often following SCS than after HMP; 10/9.7% vs. 3/2.9%, *p* = 0.041. The incidence of all other postoperative complications was comparable between groups as listed in Table 4.

### 3.5. Patient and Graft Survival

After a median follow-up of 24.2 (range 0.33–53.8) months, 1- and 3-year patient survival was 99.0% and 97.1%, respectively. The 1- and 3-year graft survival rate reached 95.6% and 91.8%, respectively. 

In the HMP group, 1- and 3-year patient survival was 100% and 99.0% and graft survival reached 97.1% and 95.2%, respectively. Primary non function (PNF) occurred in one patient following HMP. Following SCS, 1- and 3-year patient survival was 99.0% and 95.2%, whereas graft survival reached 94.2% and 88.4%, respectively. Patient and graft survival rates at 3-years post-transplant did not differ significantly between HMP and SCS (log rank, *p* = 0.185 and *p* = 0.272, respectively, Figure 3). 

Graft survival was superior in patients with immediate graft function compared to patients with DGF. This difference was statistically significant at both, one and three years post-transplant: in patients with immediate graft function, 1- and 3-year graft survival reached 100% and 96.3% compared to 90.8% and 86.7% in patients with DGF (log rank, *p* = 0.001 and *p* = 0.008, respectively, Figure 4). 

In a Cox regression analysis, recipients suffering from DGF had a 4.1-times increased risk of graft failure (HR = 4.108; 95% CI: 1.336–12.631; *p* = 0.014). One year patient survival was 100% in patients with immediate graft function and 99% in patients with DGF (log rank, *p* = 0.296). Patient survival at 3 years after transplantation was superior in cases with immediate graft function vs. DGF, but the difference did not reach statistical significance (99.1% vs. 94.9%; log rank, *p* = 0.063). 

In patients receiving a kidney after HMP, the glomerular filtration rate (GFR) at last follow-up was 44.8 mL/min/1.73 m^2^ (range 11.3–122.4) compared to 39.0 mL/min/1.73m^2^ (range 16.7–98.8) in patients after transplanting an SCS organ; (*p* = 0.177). Serum creatinine at last follow-up was 1.54 mg/dL (range 0.5–4.14) after HMP and 1.61 mg/dL (range 0.8–3.80) in patients after SCS (*p* = 0.260).

Serum creatinine and GFR at last follow-up were inferior in patients with DGF: 1.92 mg/dL (range 0.9–4.14) vs. 1.44 mg/dL (range 0.5–3.9) and 33.9 mL/min/1.73 m^2^ (range 11.3–82.0) vs. 45.9 mL/min/1.73 m^2^ (range 15.6–122.4); *p* < 0.001 for both, Figure 5. This difference was also evident in subgroup analysis (HMP vs. SCS, see Table 5). Serum creatinine and GFR at last follow-up were also inferior in patients who received an ECD kidney compared to those who received a SCD organ: 1.85 mg/dL (range 0.5–3.9) vs. 1.37 mg/dL (range 0.7–4.1) and 34.5 mL/min/1.73 m^2^ (range 15.6–122.4) vs. 51.7 mL/min/1.73 m^2^ (range 11.3–99.4); *p* < 0.001 for both.

## 4. Discussion

The present study assesses the outcome after deceased donor kidney transplantation in organs undergoing HMP compared to a propensity score matched SCS cohort. Our analysis reveals a relatively strong clinical benefit of HMP with significant lower DGF rates in both ECD and SCD kidneys. However, the novel and important finding of our study is the fact that we can operate in a back-to-base fashion, accepting SCS kidneys and connecting them to HMP as soon as the organs arrive at our center. To our knowledge, our study describes for the first time that now a similar beneficial result with HMP over SCS can be achieved with SCS and end-HMP for 6.2 hours. Boffa et al presented the POMP trial by the COPE consortium [23] fairly recently at the congress of the British Transplant Society. In their prospective trial, the combination of SCS and oxygenated end-HMP in ECD kidneys did not result in any improvement of outcomes, despite shorter CIT compared to our cohort and an HMP time of 4.7 hours. Given the fact that 6.2 hours in our cohort seemed to have a beneficial effect, it will require more investigational procedures to detect the mechanism of this very impact and, if possible, to define a certain time frame of SCS and HMP leading to a favorable outcome.

Delayed graft function is the most significant early complication after deceased donor kidney transplantation. Further to its immediate effect on patient wellbeing and hospital stay, DGF has a negative long term impact on graft and patient survival [24]. The incidence of DGF ranges from 20% to 70% and it differs significantly between regions and kidney transplant programs [25]. In several randomized controlled trials, HMP has been shown to decrease DGF rates and to boost graft survival, especially in ECD and DCD kidneys [12,26,27]. In the unmatched comparison of this trial, the DGF rate was comparable between HMP and SCS (29.8% following HMP and 32.3% in SCS). The CIT was significantly longer in the HMP group. In this regard, our study suggests that HMP may allow safe prolongation of CIT. To eliminate heterogeneity between the two cohorts, we performed propensity score matching based for key donor and recipient factors and CIT. This approach revealed that HMP was associated with a 51.5% reduction of the DGF rate (*p* = 0.001) in our cohort. While the protective effect of HMP is presumably more pronounced in ECD kidneys [4,28], we found no difference between SCD and ECD kidneys. This is in line with previously published data by Moers et al, suggesting that HMP has a beneficial effect on the short-term outcome in all types of deceased-donor kidney transplants [11]. Interestingly, the positive effect of HMP persisted despite a long CIT (16 hours). This is in contrast to data published by Kox et al who stated that organs with the shortest CIT benefitted the most, whereas the DGF rate in kidneys with over 10 hours of CIT did not differ between HMP and SCS [29]. 

Graft survival at three years reached 91.8% (95.2% following HMP and 88.4% following SCS). In our series, the difference in graft survival between HMP and SCS did not reach statistical significance. With the apparent trend, we believe that this is due to the small sample size and reference properly powered studies (e.g., the Eurotransplant trial [2]), which reported a graft survival benefit after one and three years following HMP. In our cohort, DGF was associated with a 4.1-times increased risk for graft failure, resulting in a graft survival rate of only 86.7% at three years (compared to 96.3% in non DGF kidneys). Hence, the effect of HMP in kidney transplantation seems significant and puts into question why HMP has not evolved as the new standard of care with more widely adoption. 

Moreover, HMP led to a shift towards more daytime kidney transplant procedures. The impact of the daytime of surgery on the outcomes after transplantation is contradictory. Fechner et al observed a higher incidence of complications as well as inferior graft survival following nighttime kidney transplant procedures [30]. In our cohort, outcomes were comparable between daytime and nighttime transplants. This is in line with prior published data from our center [19], as well as others [31]. Moving surgical procedures from nighttime to daytime may be particularly beneficial in times of working hour restrictions and staffing shortage. Further to this, safe prolongation of preservation times may facilitate hemodynamic stabilization in delayed kidney-liver transplantation and delayed kidney-heart transplantation or allow for pretreatment of the recipient, e.g., immunoadsorption and other desensitization strategies. 

Most studies performed in the last decade used the LifePort Kidney Transporter^®^. Further to the established HMP techniques, oxygenation during HMP has shown beneficial effect in several preclinical studies [32,33,34]. First clinical data from a double blinded, randomized, paired phase 3 trial (COPE trial) are promising, suggesting that oxygenated HMP improves graft function at one year compared to non-oxygenated HMP [35]. These findings need further confirmation, but it is reasonable to assume that oxygenated HMP will eventually be implemented and improve the outcome after kidney transplantation further.

Poor kidney function in the first week is detrimental for both the patients’ perception and the longevity of the deceased donor kidney. Hypothermic machine perfusion was found to be clearly beneficial in several randomized clinical trials. We herein provide evidence that the actual benefit of HMP in the real-world scenario might be more significant and impactful than expected. Hypothermic machine perfusion impressively halved the DGF rate compared to SCS kidneys after similar periods of CIT. For the purpose of achieving the most optimal kidney preservation, HMP should be the preferred technique. Moreover, SCS kidney transportation in the ice box is relatively cheap, but if the back-to-base approach is used and the recipient center based HMP allows to achieve a beneficial effect, this could be very attractive and still cost-effective, especially within an international long-distance exchange organization. Future studies should emphasize on the impact of different durations of CIT accompanied by short periods of HMP to detect possible “rescue” time frames to consider even short periods of end ischemic HMP, for example, 2–3 hours only, crucial for optimal outcome after deceased donor kidney transplantation.

## Figures and Tables

**Figure 1 jcm-09-02311-f001:**
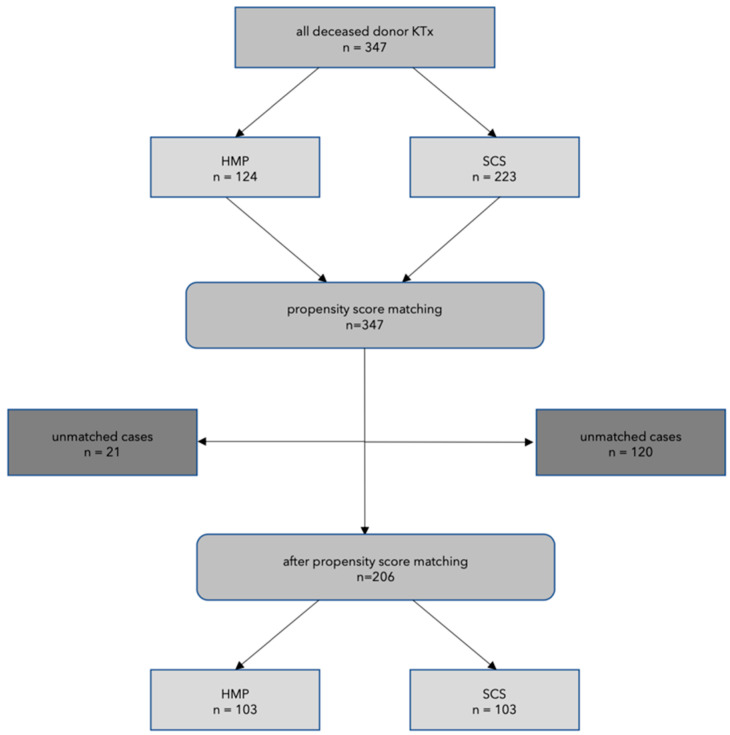
Flow chart of patient selection. HMP: hypothermic machine perfusion, SCS: static cold storage.

**Figure 2 jcm-09-02311-f002:**
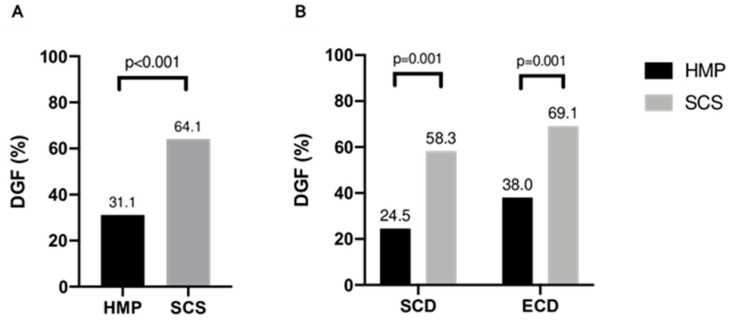
(**A**) Delayed graft function rate of hypothermically perfused deceased donor kidneys compared to kidneys statically stored on ice; (**B**) delayed graft function rate of kidneys after hypothermic machine perfusion compared to organs stored on ice stratified for standard and extended criteria donors.

**Figure 3 jcm-09-02311-f003:**
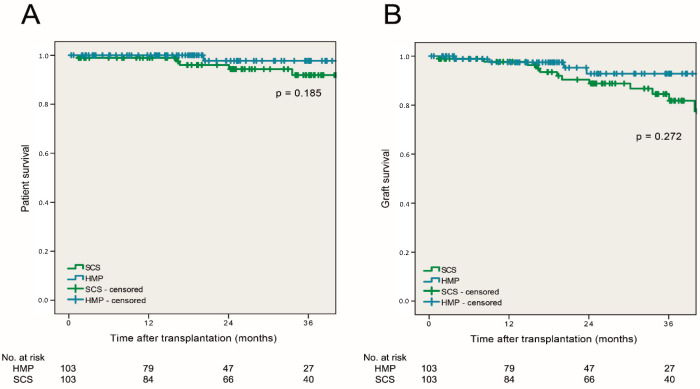
(**A**) Estimated 3-year patient and (**B**) 3-year graft survival rates after transplantation of HMP and SCS kidneys.

**Figure 4 jcm-09-02311-f004:**
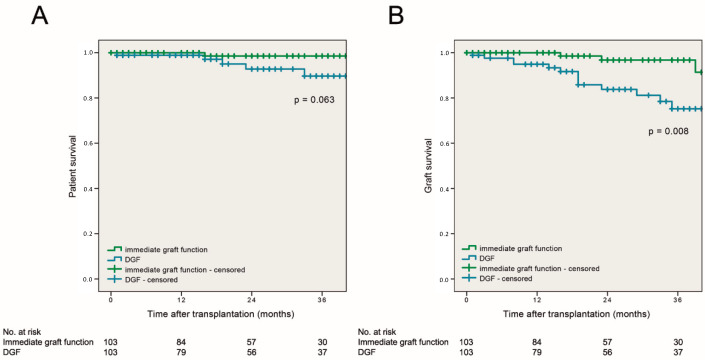
(**A**) Estimated 3-year patient and (**B**) 3-year graft survival rates of patients stratified for immediate graft function and delayed graft function.

**Figure 5 jcm-09-02311-f005:**
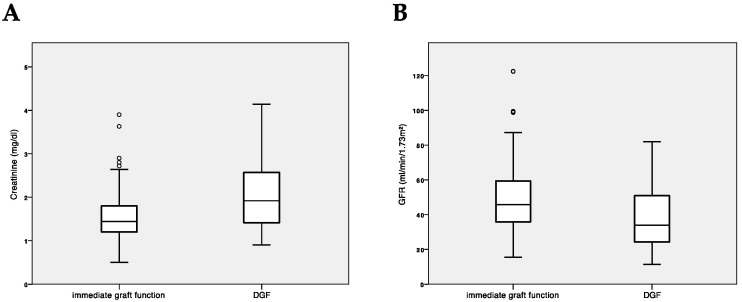
(**A**) Median serum creatinine and (**B**) estimated glomerular filtration rate (MDRD) after a median follow-up of 24 months for patients with immediate and delayed graft function. ° statistical outliers.

**Table 1 jcm-09-02311-t001:** Baseline characteristics of the unmatched kidney transplant cohort (*n* = 347).

	HMP, *n* = 124	SCS, *n* = 223	*p*-Value	SMD
CIT (hours) *	17.5 (± 5.1)	13.3 (± 4.6)	*p* < 0.001	0.865
ECD (*n*, %)	61 (49.2%)	95 (42.6%)	*p* = 0.237	0.108
SCD (*n*, %)	63 (50.8%)	128 (57.4%)	*p* = 0.237	0.108
Gender MM (*n*, %)	66 (53.2%)	110 (49.3%)	*p* = 0.486	0.064
CMV MM (*n*, %)	21 (16.9%)	38 (17.1%)	*p* = 0.966	0.004
Re-TX (*n*, %)	31 (25.0%)	44 (19.7%)	*p* = 0.253	0.103
ET-Senior (*n*, %)	25 (20.2%)	43 (19.3%)	*p* = 0.843	0.018

SMD: standardized mean difference, CIT: cold ischemia time, ECD: extended criteria donor, SCD: standard criteria donor, MM: mismatch, TX: transplantation, ET: Eurotransplant. * values are mean (SD).

**Table 2 jcm-09-02311-t002:** Baseline characteristics of the propensity-score matched cohort (*n* = 206).

	HMP, *n* = 103	SCS, *n* = 103	SMD
CIT (hours) *	16.32 (± 4.3)	15.97 (± 4.4)	0.059
ECD (*n*, %)	50 (48.5%)	55 (53.4%)	0.080
SCD (*n*, %)	53 (51.5%)	48 (46.6%)	0.080
Gender MM (*n*, %)	54 (52.4%)	49 (47.6%)	0.078
CMV MM (*n*, %)	17 (16.5%)	13 (12.6%)	0.092
Re-TX (*n*, %)	23 (22.3%)	25 (24.3%)	0.019
ET-Senior (*n*, %)	19 (18.4%)	16 (15.5%)	0.062

SMD: standardized mean difference, CIT: cold ischemia time, ECD: extended criteria donor, SCD: standard criteria donor, MM: mismatch, CMV: cytomegalovirus, Re-TX: re-transplantation, ET: Eurotransplant. * values are mean (SD).

**Table 3 jcm-09-02311-t003:** Donor and recipient demographics and transplant factors stratified by HMP and SCS.

	HMP, *n* = 103	SCS, *n* = 103
Recipient age, median (range)	57 (28–79)	58 (26–76)
Recipient male gender, *n* (%)	74 (71.8%)	71 (68.9%)
Recipient BMI kg/m^2^, mean ± SD	25.5 ± 4.6	26.4 ± 4.2
Prior TX, *n* (%)	23 (22.3%)	25 24.3%
Double kidney TX, *n* (%)	6 (5.8%)	9 (8.7%)
Donor age, median (range)	55 (18–82)	55 (18–84)
Donor male gender, *n* (%)	52 (50.5%)	64 (62.1%)
Donor BMI kg/m^2^, mean ± SD	26.8 ± 5.2	27.2 ± 4.4
Extended criteria donor, *n* (%)	50 (48.5%)	55 (53.4%)
Kidney donor risk index (KDRI), mean ± SD	1.29 ± 0.42	1.31 ± 0.42
Kidney donor profile index (KDPI), mean ± SD	66.7 ± 23.9	67.3 ± 25.3
Cause of end stage renal disease		
Glomerulonephritis	37 (35.9%)	40 (38.8%)
Diabetic nephropathy	19 (18.5%)	16 (15.5%)
Hereditary renal disease	12 (11.7)	15 (14.6%)
Vascular nephropathy	12 (11.7%)	13 (12.6%)
Others	23 (22.3%)	19 (18.4%)
Cold ischemia time in hours, mean ± SD	16.32 ± 4.3	15.97 ± 4.4
Anastomosis time in minutes, mean ± SD	33 ± 10	31 ± 9
HLA A mm, mean ± SD	1.02 ± 0.61	0.98 ± 0.69
0 and 1	83 (80.6%)	80 (77.7%)
2	20 (19.4%)	22 (21.3%)
HLA B mm, mean ± SD	1.29 ± 0.69	1.15 ± 0.66
0 and 1	59 (57.3%)	72 (69.9%)
2	44 (42.7%)	31 (30.1%)
HLA DR mm, mean ± SD	1.20 ± 0.63	1.02 ± 0.71
0 and 1	70 (68.0%)	76 (73.8%)
2	33 (32.0%)	27 (26.2%)

BMI: body mass index, SD: standard deviation, TX: transplantation, HLA: human leukocyte antigen.

**Table 4 jcm-09-02311-t004:** Postoperative complications of the propensity-score matched cohort.

	HMP, *n* = 103	SCS, *n* = 103	*p*-Value
overall complications	27 (26.2%)	29 (28.2%)	0.438
hematoma	6 (5.8%)	8 (7.8%)	0.392
lymphocele	6 (5.8%)	13 (12.6%)	0.074
urological complications	8 (7.8%)	7 (6.8%)	0.500
urinary leakage	4 (3.9%)	3 (2.9%)	
ureteral stenosis	2 (1.9%)	4 (3.9%)	
vesicoureteral reflux	2 (1.9%)	0	
wound infections	3 (2.9%)	10 (9.7%)	0.041
vascular complications	4 (3.8%)	3 (2.9%)	0.500
arterial stenosis	1 (1.0%)	2 (1.9%)	
pseudoaneurysm	1 (1.0%)	0	

**Table 5 jcm-09-02311-t005:** Subgroup analysis of serum creatinine and eGFR in patients with immediate graft function and DGF.

	DGF	No DGF	*p*-Value
HMP			
serum creatinine	2.0 (0.9–4.1)	1.42 (0.5–3.9)	0.001
GFR	32.9 (11.3–76.3)	46.8 (15.6–122.4)	0.004
SCS			
serum creatinine	1.84 (0.92–3.8)	1.5 (0.8–3.63)	0.009
GFR	33.9 (16.7–82.0)	42.2 (19.6–98.8)	0.015

GFR: glomerular filtration rate; values are median (range).

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
