# Peer review of "A Retrospective Propensity Score Matched Analysis Reveals Superiority of Hypothermic Machine Perfusion over Static Cold Storage in Deceased Donor Kidney Transplantation"

_jcm, 2020, doi:10.3390/jcm9072311_

Round 1

Reviewer 1 Report

This is a single center study showing propensity scored superiority of kidneys undergoing hypothermic machine perfusion over static cold storage  at a mean period of 6.2 hours.  The advantages of hypothermic machine perfusion are well known in cases of prolonged cold ischemic times. there needs to be more emphasis on why this technique is important at lower cold ischemic times.

Author Response

This is a single center study showing propensity scored superiority of kidneys undergoing hypothermic machine perfusion over static cold storage  at a mean period of 6.2 hours.  The advantages of hypothermic machine perfusion are well known in cases of prolonged cold ischemic times. there needs to be more emphasis on why this technique is important at lower cold ischemic times.

Answer: Thank you for your positive comment.

We greatly appreciate your suggestion to emphasize durations of cold ischemia and mentioned this in the discussion section. Our results indicate that we can achieve a similar beneficial effect with end-HMP after a certain period of SCS. This back-to-base approach is very attractive and cost-effective. You are completely right, now we need to focus/investigate on the specific time frame of SCS and HMP were we can still observe this beneficial effect. There is only scarce data available focusing on this specific topic. As we already mentioned in the discussion section (line 327 onwards), to our knowledge, this is the first study which describes a similar beneficial effect of end-HMP as compared to HMP only. In our analysis, an average of 6 hours End-HMP after a period of 10 hours of SCS seems to be beneficial. In the recently published data of the POMP trial, with 4.7 hours of End-HMP after 8.5 hours of SCS, no beneficial effect could be observed

Regrettably we do not have sufficient data at the moment to support such presentation of data. To figure out the importance of HMP in respect to variable SCS, prospective trials, including centers applying the back-to-base approach, are desirable (see line 395 onwards)

Reviewer 2 Report

Gasteiger S at el. described a retrospective propensity score matched analysis reveals superiority of hypothermic machine perfusion over static cold storage in deceased donor kidney transplantation.

I congratulate the authors for their results. This study presents the results of a clinical study comparing kidney storage with classical cold storage versus hypothermic machine perfusion. It includes a reasonably large cohort of patients, although not randomized. I have some questions:

Did the authors have paired kidneys to compare 1:1 HMP versus SCS?

What day post op was the tacrolimus introduced? Why the high prograf level

Did the patients with DGF received any extra induction?

Any rejection episodes?

Any difference in the creatinine, rejection and eGFR between SCS group (with DGF and no DGF) and the HMP group (with DGF and no DGF)

Any vascular thrombosis and/or primary non-functioning

How were the patients with DGF treated?

Did the authors add any medication to the HMP to improve IRR index?

How long were the patients on DGF and how they adjusted immune meds? Did the authors perform biopsies in patient with DGF?

Who puts the kidney on the machine? Who monitors the kidney? What is the determining factor to determine when the kidney is ready for transplantation?

Author Response

Comments and Suggestions for Authors

Gasteiger S at el. described a retrospective propensity score matched analysis reveals superiority of hypothermic machine perfusion over static cold storage in deceased donor kidney transplantation.

I congratulate the authors for their results. This study presents the results of a clinical study comparing kidney storage with classical cold storage versus hypothermic machine perfusion. It includes a reasonably large cohort of patients, although not randomized. I have some questions:

Answer: Thank you very much for your positive comment.

Did the authors have paired kidneys to compare 1:1 HMP versus SCS?

Answer: This is an important comment: Indeed, we had kidney pairs in our analysis: 32 pairs, 64 kidneys

27 pairs (54 kidneys) underwent the same method of preservation; both either SCS or HMP resulting in 16 DGF in SCS and 16 no DGF in SCS and 10 DGF in HMP and 12 no DGF in HMP

5 pairs (10 kidneys) underwent different preservation methods; one SCS and one HMP resulting in 1 DGF in SCS and 4 no DGF in SCS and no DGF in HMP.

Due to the low number of cases and the retrospective nature of our study, and the insignificant differences, we aimed to avoid too much granularity of data.

For future prospective trials the paired approach of analysis would be the preferred way.

What day post op was the tacrolimus introduced? Why the high prograf level

Answer: Tacrolimus (prograf formulation) was started immediately after transplantation on the first post op day. Trough levels 8-10 ng/ml for the first 3 months are our institutional standard for the first kidney transplant in an adult; 8 ng/ml thereafter. In the case of retransplant, we aim for 10ng/ml in the first 3 months (see line 142 onwards).

Did the patients with DGF received any extra induction?

Answer: Induction treatment (antithymocyte globulin) is only administered in retransplant-cases and started during the process of transplant surgery already. No additional treatment is given in cases of DGF. No standardized protocol biopsies are performed during the delayed functional period (see line 149).

Any rejection episodes?

Answer: Rejections were seen in both groups. 14/124 (11.29%) in the HMP group (overall, before matching) were treated for an episode of acute rejection; 22/223 (9.9%) in the SCS group (overall, before matching). All of them could be treated successfully with 3 bolus-doses of steroids (500 mg per day).

Any difference in the creatinine, rejection and eGFR between SCS group (with DGF and no DGF) and the HMP group (with DGF and no DGF)

Answer: Rejection episodes were not significantly different between the groups SCS/HMP and DGF/no DFG. Therefore, rejection was not considered as a parameter for the propensity score matched analysis.

The difference in eGFR and serum creatinine between SCS group and HMP group did not reach statistical significance (see line 297). Although the observed difference among patients with DGF and no DGF (line 301) was also evident in subgroup analysis (SCS and HMP). As suggested, we have added an additional table 5 (see line 319)

Any vascular thrombosis and/or primary non-functioning

Answer: Vascular thrombosis did not occur in any of our patients, we observed in total 3 vascular stenosis and 1 pseudoaneurysm, all of them were successfully treated by our interventional radiologists (see table 4, line 258).

There was one primary non function which is included in the graft survival analysis; we amended the results section and added this event (line 264).

How were the patients with DGF treated?

Answer: There was no specific treatment for patients with DGF; neither changes in immunosuppression, nor protocol biopsies (see line 149)

Did the authors add any medication to the HMP to improve IRR index?

Answer: No, nothing was added as there are not any licensed drugs available to treat the kidney during HMP.

How long were the patients on DGF and how they adjusted immune meds? Did the authors perform biopsies in patient with DGF?

Answer: No additional treatment was given in cases of DGF; no standardized protocol biopsies were performed during the delayed functional period according to our clinical SOP for deceased donor kidney transplantation (line 142 onwards). The kidney function in the DGF cases took about 14 to 17 days to recover fully.

Who puts the kidney on the machine? Who monitors the kidney? What is the determining factor to determine when the kidney is ready for transplantation?

Answer: In our institution, the procedures you mentioned are exclusively performed by surgeons (consultants and trainees). The kidney is ready for transplantation when the theatre capacity is available and, in addition, when the intra-renal resistance is 0.3 or less.

Reviewer 3 Report

The present study aimed at retrospectively compare the outcomes of kidney transplantation (KT) cases with kidney grafts alternatively managed in the pre-transplant period either with static cold storage (n=233) or hypothermic machine perfusion (n=124). Double graft KT cases were included while combined transplants or transplants using graft from donation after cardiac death (DCD) were excluded. The stated rationale was that the data on HMP regarding its potential protective effect against cold ischemia injury and its potential reconditioning effect of marginal grafts, are heterogeneous.  However, the use of HMP in the present study population was actually targeted to improve the outcome but rather to prolong the cold ischemia time (CIT) because the KT could not be performed in urgent setting for logistic reasons. As a matter of fact, in the baseline confront of the HMP and SCS group, only the CIT duration was significantly different, while the prevalence of marginal grafts (ECD), elderly recipients or retransplant cases were comparable between the groups. Under this perspective, the present study replicates previous studies of greater significance, including randomized control trails and national registry reports (Wang W, Xie D, Hu X, Yin H, Liu H, Zhang X. Effect of Hypothermic Machine Perfusion on the Preservation of Kidneys Donated After Cardiac Death: A Single-Center, Randomized, Controlled Trial. Artif Organs. 2017;41(8):753-758; Kox J, Moers C, Monbaliu D, et al. The Benefits of Hypothermic Machine Preservation and Short Cold Ischemia Times in Deceased Donor Kidneys. Transplantation. 2018;102(8):1344-1350; 2.            Peng P, Ding Z, He Y, Zhang J, Wang X, Yang Z. Hypothermic Machine Perfusion Versus Static Cold Storage in Deceased Donor Kidney Transplantation: A Systematic Review and Meta-Analysis of Randomized Controlled Trials. Artif Organs. 2019 May;43(5):478-489)

Nonetheless the numerosity of the study population and the robust statistical analysis are of value.

Methods

- Which definition for ECD graft was used? A prevalence of nearly 50% in the whole population seems quite high

- Which criteria were used for double graft KT?

- Was any kidney discarded at the end of HMP due to poor hemodynamic parameters?

- Please describe how postoperative complications were defined and considered.

- line 105 "The effect of HMP on the incidence of DGF was modelled with the conditional logistic regression analysis". Please better clarify the type of analysis used.

Results

- The granularity of the data regarding the clinical characteristics of donors and recipients is too low. For example, donor age, HLA compatibility, donor terminal creatinine, dyalisis duration, single vs double kidney grafts, warm ischemia time, etc...

- a DGF prevalence of 47.6% in the overall population and reaching 64.1% in the SCS groups, seems extremely high, even considering the exclusion of DCD grafts and a not extremely long CIT (mean 16 hours)

- Daytime vs. nighttime procedures confront is off-topic as presented.If the Authors wants to investigate the advantage of using HMP to postpone the procedure from night-tine to day-time, then a comparison between SCS-night-time vs HMP-day-time should have better being performed

- The analysis of the impact of DGF on graft survival and function in the whole study population is unnecessary and off-topic.

Discussion

The claimed strength novelty of the study was of showing the delayed HMP after a preliminary period of SCS is still beneficial. The result is surely interesting thanks to the numerosity of the HMP group, but surely not new (see the discussion of Adani GL, Pravisani R, Crestale S, et al. Effects of Delayed Hypothermic Machine Perfusion on Kidney Grafts with a Preliminary Period of Static Cold Storage and a Total Cold Ischemia Time of Over 24 Hours. Ann Transplant. 2020;25:e918997)

Author Response

The present study aimed at retrospectively compare the outcomes of kidney transplantation (KT) cases with kidney grafts alternatively managed in the pre-transplant period either with static cold storage (n=233) or hypothermic machine perfusion (n=124). Double graft KT cases were included while combined transplants or transplants using graft from donation after cardiac death (DCD) were excluded. The stated rationale was that the data on HMP regarding its potential protective effect against cold ischemia injury and its potential reconditioning effect of marginal grafts, are heterogeneous.  However, the use of HMP in the present study population was actually targeted to improve the outcome but rather to prolong the cold ischemia time (CIT) because the KT could not be performed in urgent setting for logistic reasons. As a matter of fact, in the baseline confront of the HMP and SCS group, only the CIT duration was significantly different, while the prevalence of marginal grafts (ECD), elderly recipients or retransplant cases were comparable between the groups. Under this perspective, the present study replicates previous studies of greater significance, including randomized control trails and national registry reports (Wang W, Xie D, Hu X, Yin H, Liu H, Zhang X. Effect of Hypothermic Machine Perfusion on the Preservation of Kidneys Donated After Cardiac Death: A Single-Center, Randomized, Controlled Trial. Artif Organs. 2017;41(8):753-758; Kox J, Moers C, Monbaliu D, et al. The Benefits of Hypothermic Machine Preservation and Short Cold Ischemia Times in Deceased Donor Kidneys. Transplantation. 2018;102(8):1344-1350; 2.            Peng P, Ding Z, He Y, Zhang J, Wang X, Yang Z. Hypothermic Machine Perfusion Versus Static Cold Storage in Deceased Donor Kidney Transplantation: A Systematic Review and Meta-Analysis of Randomized Controlled Trials. Artif Organs. 2019 May;43(5):478-489)

Nonetheless the numerosity of the study population and the robust statistical analysis are of value.

Answer: Thank you very much for your positive comments.

Methods

- Which definition for ECD graft was used? A prevalence of nearly 50% in the whole population seems quite high

Answer: Extended criteria donors (ECD) were defined as age ≥ 60 years, or a donor over the age of 50 with two of the following: a history of high blood pressure, a creatinine greater than or equal to 1.5, or death resulting from cardiovascular accident as defined in 2002 by Port et. al.: Port FK, Bragg-Gresham JL, Metzger RA, Dykstra DM, Gillespie BW, Young EW, et al. Donor characteristics associated with reduced graft survival: an approach to expanding the pool of kidney donors. Transplantation. 2002;74(9):1281-6.

We have amended the definition in the material and methods section (line 138).

- Which criteria were used for double graft KT?

Answer: We apply the Remuzzi score and also the donor eGFR to decide on single or dual kidney transplantation – subject to recipient’s performance.

Donor eGFR ≥60 single transplant; 30-60 dual transplant; ≤30 discard

Remuzzi 0-4 single transplant, 5-6 dual transplant ≥7 discard

- Was any kidney discarded at the end of HMP due to poor hemodynamic parameters?

Answer: No, the current analysis includes transplanted kidneys only. Considering all data and donor factors available, we always have a close look on the development of the intra-renal resistance: DBD kidney needs to be at 0.3 or below; DCD at 0.4 or below  according to our institutional standard.

- Please describe how postoperative complications were defined and considered.

Answer: Complications were defined as any unexpected event not intrinsic to the procedure, which occurred within the period of follow-up (line 250 onwards).

- line 105 "The effect of HMP on the incidence of DGF was modelled with the conditional logistic regression analysis". Please better clarify the type of analysis used.

Answer: Conditional logistic regression (CLR) was used due the fact that we analyzed our cohort using propensity score matching. CLR is a specialized type of logistic regression usually employed when case subjects with a particular condition or attribute (HMP for example in our study) are each matched with X control subjects (SCS for example) without the condition. The main field of applying CLR is for observational studies and epidemiology in particular.

Results

- The granularity of the data regarding the clinical characteristics of donors and recipients is too low. For example, donor age, HLA compatibility, donor terminal creatinine, dyalisis duration, single vs double kidney grafts, warm ischemia time, etc...

Answer: Thank you for this suggestion. We added an additional table 3 to give an overview of donor and recipient demographics as well as transplant factors for the propensity score matched cohort (line 215).

- a DGF prevalence of 47.6% in the overall population and reaching 64.1% in the SCS groups, seems extremely high, even considering the exclusion of DCD grafts and a not extremely long CIT (mean 16 hours)

Answer: You are absolutely right – this is a very high percentage of DGF. However, it emphasizes the importance of HMP and shows the number of cases in the propensity score matched SCS group. Matching is and was important to highlight events under the same circumstances – as far as we can create similar conditions for a cohort which needs to be compared in a retrospective analysis.

The overall DGF-rate of the unmatched cohort is comparable to published and well-accepted data with approximately 30% (line 146 onwards).

- Daytime vs. nighttime procedures confront is off-topic as presented. If the Authors wants to investigate the advantage of using HMP to postpone the procedure from night-tine to day-time, then a comparison between SCS-night-time vs HMP-day-time should have better being performed

Answer: We greatly appreciate your comment. Our intent was to underline the real-world clinical-reality scenario.

- The analysis of the impact of DGF on graft survival and function in the whole study population is unnecessary and off-topic.

Answer: Thank you for your comment. We have included graft and patient survival in our analysis as the event DGF itself is one of the predictors for them.

Discussion

The claimed strength novelty of the study was of showing the delayed HMP after a preliminary period of SCS is still beneficial. The result is surely interesting thanks to the numerosity of the HMP group, but surely not new (see the discussion of Adani GL, Pravisani R, Crestale S, et al. Effects of Delayed Hypothermic Machine Perfusion on Kidney Grafts with a Preliminary Period of Static Cold Storage and a Total Cold Ischemia Time of Over 24 Hours. Ann Transplant. 2020;25:e918997)

Answer: Thank you for your comment. You are absolutely right that plenty of data about HMP and SCS had been and have been studied and published already. Our goal was to investigate and to confirm published trial and registry date in our single center as our very own quality control to justify the application of HMP according to institutional policy to really achieve a local impact.

Fortunately, we were able to demonstrate that within our own remit when receiving kidneys, which are allocated to us by Eurotransplant, those kidneys that are pumped on HMP have better outcomes than the SCS group. The novel finding in our study is the fact that the back-to-base approach in HMP kidney preservation brings a similar beneficial result with HMP over SCS as the immediate approach has shown before (this is significantly different to the paper published by the Italian research group you quoted with n=21 kidney allografts).

Round 2

Reviewer 2 Report

This paper is going to help transplant centers in the use of machine perfusion